# Quantifying impacts of stony coral tissue loss disease on corals in Southeast Florida through surveys and 3D photogrammetry

**Ian R. Combs**[¤a]*, **Michael S. Studivan**[¤b], **Ryan J. Eckert**, **Joshua D. Voss**

Department of Biological Sciences, Harbor Branch Oceanographic Institute, Florida Atlantic University, Fort Pierce, Florida, United States of America

¤a Current address: Elizabeth Moore International Center for Coral Reef Research & Restoration, Mote Marine Laboratory, Summerland Key, Sarasota, Florida, United States of America
¤b Current address: Cooperative Institute for Marine and Atmospheric Studies, University of Miami, Miami, Florida, United States of America
* combsi9892@gmail.com

**Data Availability Statement:** Model generation protocol, analysis scripts, and documentation are available in a GitHub repository [49], doi:10.5281/zenodo.4728238. Disease prevalence observations

## Abstract

Since 2014, stony coral tissue loss disease (SCTLD) has contributed to substantial declines of reef-building corals in Florida. The emergence of this disease, which impacts over 20 scleractinian coral species, has generated a need for widespread reef monitoring and the implementation of novel survey and disease mitigation strategies. This study paired SCTLD prevalence assessments with colony-level monitoring to help improve understanding of disease dynamics on both individual coral colonies and at reef-wide scales. Benthic surveys were conducted throughout the northern Florida Reef Tract to monitor the presence/absence of disease, disease prevalence, and coral species affected by SCTLD. Observed SCTLD prevalence was lower in Jupiter and Palm Beach than in Lauderdale-by-the-Sea or St. Lucie Reef, but there were no significant changes in prevalence over time. To assess colony-level impacts of the disease, we optimized a low-cost, rapid 3D photogrammetry technique to fate-track infected *Montastraea cavernosa* coral colonies over four time points spanning nearly four months. Total colony area and healthy tissue area on fate-tracked colonies decreased significantly over time. However disease lesion area did not decrease over time and was not correlated with total colony area. Taken together these results suggest that targeted intervention efforts on larger colonies may maximize preservation of coral cover. Traditional coral surveys combined with 3D photogrammetry can provide greater insights into the spatiotemporal dynamics and impacts of coral diseases on individual colonies and coral communities than surveys or visual estimates of disease progression alone.

## Introduction

Coral cover in the Tropical Western Atlantic (TWA) has declined over the last four decades [1,2], and coral diseases have been identified as one major driver of widespread coral decline throughout the region [3]. In the 1990s, white band disease dramatically reduced coral cover

can be found in the S1 Dataset. Surface area measurements from the fate-tracked coral colonies can be found in the S2 Dataset. Surface area and error measurements for the three prism shapes can be found in S3 Dataset.

**Funding:** Funding for this research was awarded to Dr. Joshua D Voss from the Florida Department of Environmental Protection (awards B430E1 and B55008), the Environmental Protection Agency (South Florida Geographic Initiative award X7 00D667-17. Mr. Jeff Beal co-PI), and the NOAA Coral Reef Conservation Program (award NA16NOS4820052). Additional funding was awarded to Mr. Ian R Combs by the Harbor Branch Oceanographic Institute Foundation through the Indian River Lagoon Graduate Research Fellowship.

**Competing interests:** The authors have declared that no competing interests exist.

of *Acropora cervicornis* and *Acropora palmata* by 95% in the TWA [4]. At the same time, in the Florida Keys, white pox was responsible for up to a 70% reduction in *A. palmata* cover [5]. Additionally, increased disease prevalence following a coral bleaching event in 2005 caused a 60% decline in coral cover throughout the U.S. Virgin Islands [6].

Coral diseases, regardless of their host specificity, have contributed to the decline of many integral reef-building scleractinian species. While white pox, white band, and acute *Montipora* white syndrome are genus-specific, other diseases have a broad, even pan-oceanic host range [6–8]. For example, white plague type II affects 17 species of scleractinian coral across multiple genera [9]. Widespread decreases in coral cover from diseases result in an ecological shift from diverse, coral-dominated communities to more homogenous, algal-dominated communities [10,11]. Such shifts have the potential to create long-term changes in fish assemblages and fishery yields [12] as well as the loss of key ecosystem services such as fisheries habitat [13], coastal wave protection [14] and nutrient cycling [15] that can persist long after a coral disease event subsides.

Since 2014, the Florida Reef Tract (FRT) has experienced an ongoing outbreak of a newly-described coral disease responsible for widespread coral mortality throughout the TWA. Stony coral tissue loss disease (SCTLD) is characterized as a highly virulent disease that affects over 20 species of scleractinian corals in the TWA [16]. SCTLD first appeared in the summer of 2014 following the dredging of Government Cut in Miami-Dade County [17,18]. In subsequent years, reports of SCTLD infections have increased and spread from Miami-Dade County along the Florida Reef Tract (FRT) and into the wider TWA. To date, SCTLD has spread north to the northern terminus of the FRT in Martin County and south past the Marquesas Keys in Monroe County, with additional outbreaks observed in at least twelve territories throughout the TWA [19,20]. Spatial epidemiological modeling revealed that SCTLD is highly contagious with significant disease clusters as wide-spread as 140 km confirming the severity of this outbreak [21].

SCTLD manifests as lesions of necrotic tissue that spread across a colony, leaving behind denuded coral skeleton (Fig 1) [16]. SCTLD is histologically distinct from white plague, exhibiting a fast-acting, liquefactive necrosis after lesions develop deep within coral tissue and progress to the colony surface [22]. Across multiple host coral species, SCTLD-affected colonies demonstrate altered microbial communities relative to their apparently healthy counterparts [23–25]. While both microbial culture-based and sequencing studies are ongoing, no pathogen for SCTLD has been identified to date. Various intervention methods have been proposed and trialed to attempt to reduce immediate loss of coral cover due to colony mortality, including probiotic treatments, physical interventions, and various topical treatments [26]. The most popular has been a combination of physical intervention by trenching a firebreak around the infected tissue and covering the infected tissue with a topical antibiotic application [27]. Regardless of the method, the various proposed techniques require individual evaluation and continued monitoring to determine success.

Traditionally, coral disease monitoring has relied on a combination of benthic survey methods and individual colony monitoring, the latter often based on estimations of disease area, *in situ* linear measurements, or planar photography [28,29]. However, these techniques may incorporate a substantial observer bias or uncertainties associated with 2D estimations of 3D surfaces [30]. Underwater photogrammetry and 3D modeling of coral colonies is an emerging technique that offers the potential to enhance both the accuracy and speed of data collection. Structure-from-motion (SfM) photogrammetry derives 3D structure from a series of overlapping images, similar to established stereoscopic methods. Rather than requiring known *a priori* positions, SfM orientation is resolved based on common features extracted from overlapping images [31]. Conveniently, this can be done using a moving sensor, or from still images generated by an individual with a camera. Photogrammetry has been used in a number of coral reef

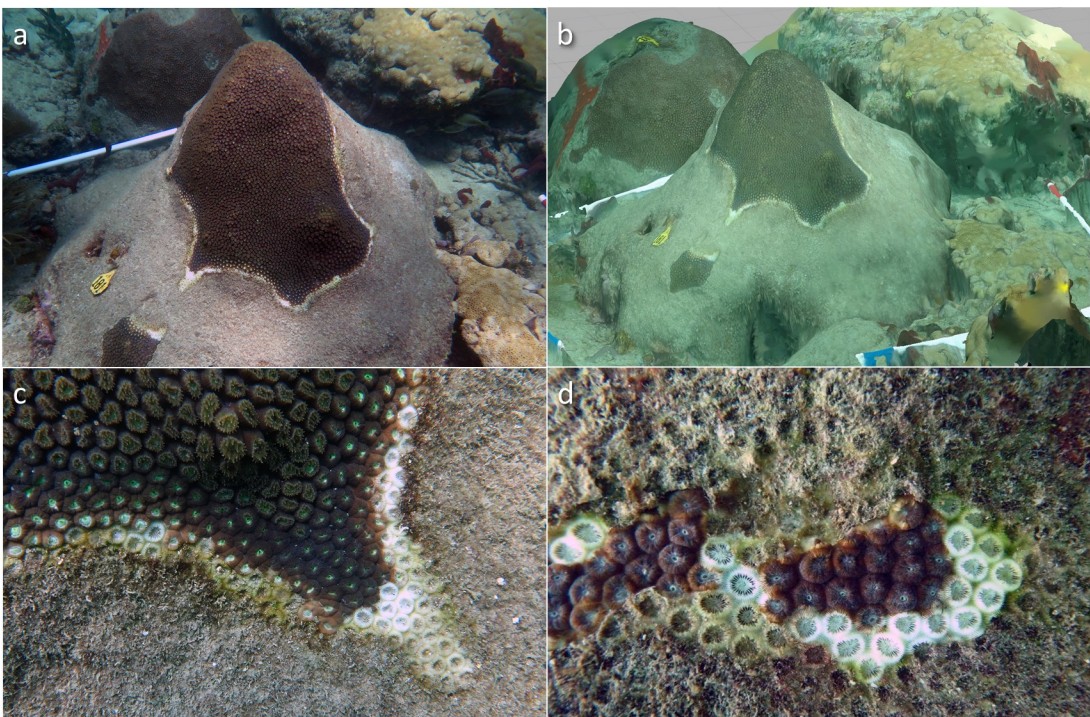

**Fig 1. SCTLD lesions on a colony of *Montastraea cavernosa*.** (a) Fate-tracked, SCTLD-infected *Montastraea cavernosa* with (b) rendered 3D model, (c) characteristic disease lesion, and (d) necrotic tissue.

applications including habitat characterizations [32–34], surface and volume measurements [35,36], and growth measurements [37,38]. SfM photogrammetry was first implemented on coral colonies using 10–20 overlapping photographs to generate 3D models [39]. As technology has progressed and computational power has become increasingly affordable, SfM photogrammetry techniques for the purpose of assessing various coral metrics have been refined and improved upon [35,38]. Changes in image capture and model reconstruction have improved measurement accuracy and precision, especially in regards to more established surface area methods such as the foil-wrapping method [30,35,38,40,41]. As such, SfM photogrammetry has emerged as a powerful and commonly-used tool for coral reef researchers.

In this study, we adapted a SfM photogrammetry technique [41] to track disease progression in fate-tracked *Montastraea cavernosa* colonies in Southeast Florida to gain insight into the colony-level dynamics of SCTLD. The study was designed to provide insight into colony- and community-level dynamics of this poorly understood disease through a combination of roving diver surveys and colony fate-tracking using SfM photogrammetry. The ultimate goal of this work is to increase widespread application of this and similar techniques to improve the design, implementation, and success of coral disease intervention, mitigation, and management strategies.

## Methods

### Disease prevalence surveys

Four locations across the northern Florida Reef Tract (NFRT) were selected for disease prevalence surveys: St. Lucie Reef (SLR), Jupiter (JUP), Palm Beach (PB), and Lauderdale-by-the-Sea (LBTS, Fig 2). The work conducted at SLR was done so under the St. Lucie Inlet State Park Permit 06261715, issued by St. Lucie Inlet State Park and FWC SAL-17-1960-SCRP, issued by

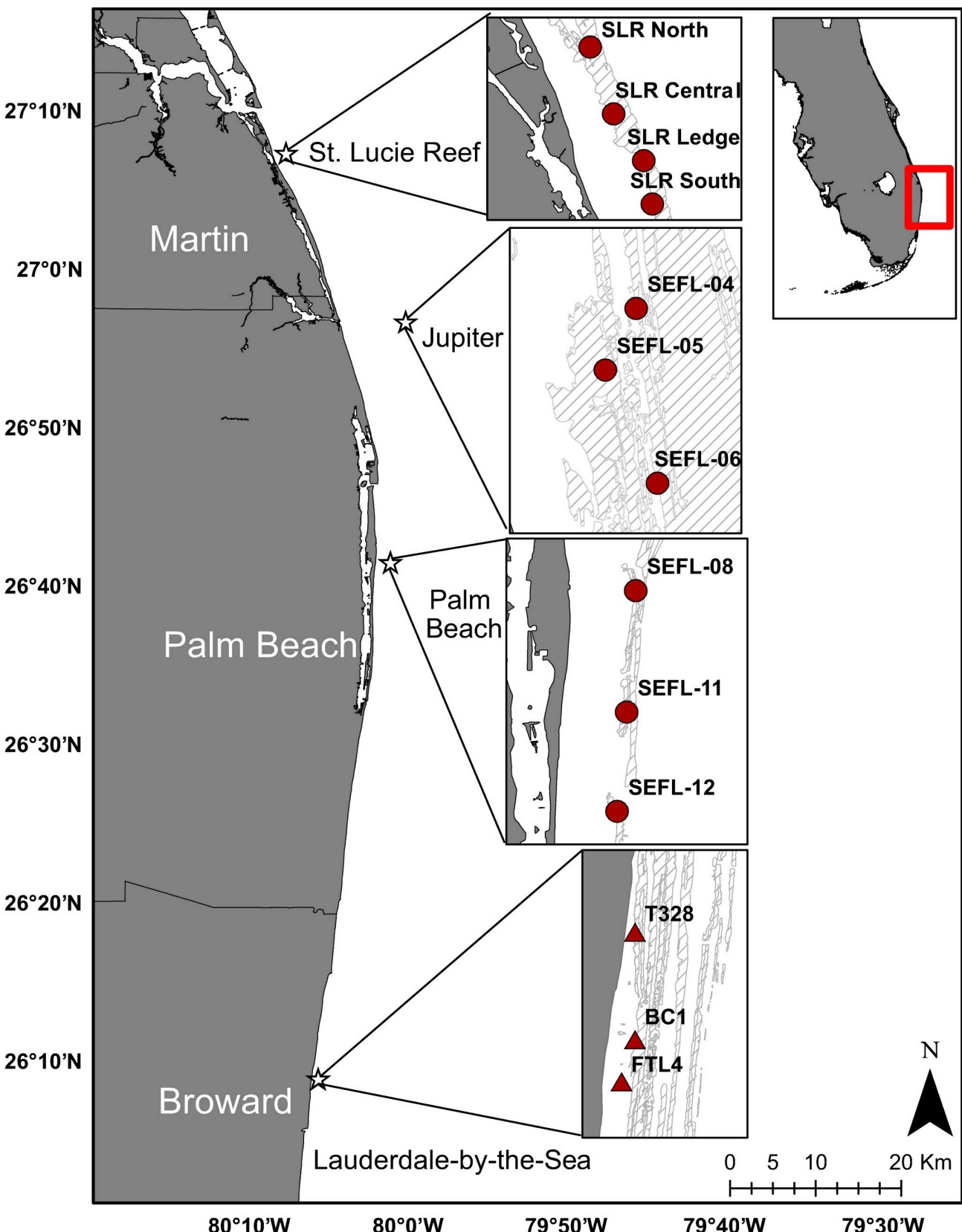

**Fig 2. Map of study locations throughout the Northern Florida Reef Tract Florida.** Red circles indicate roving diver survey sites and red triangles indicate sites where both roving diver surveys and coral fate-tracking occurred (Florida boundary and benthic hard bottom shapefile source: Florida Fish and Wildlife Conservation Commission-Fish and Wildlife Research Institute).

Florida Fish and Wildlife Conservation Commission. The work conducted at JUP, PB, and LBTS was conducted under FWC SAL-17-1960-SCRP, issued by Florida Fish and Wildlife Conservation Commission. Following Hurricane Irma in September 2017, a rapid-response damage and disease survey effort was conducted throughout Southeast Florida [42]. Data from these initial surveys were used to select the sites for the present study based on occurrence of coral communities and prevalence of SCTLD (Fig 2).

Roving diver disease surveys were conducted approximately monthly from November 2017 to June 2019. These surveys were designed to assess the greatest reef area possible, quantifying disease prevalence over an estimated range of 100–2000 $m^2$ per survey dependent on underwater visibility. SCUBA divers swam for 20 min and recorded the species and disease status of every living coral colony ≥10 cm in diameter, and SCTLD abundance and prevalence were calculated from raw counts data. To assess multivariate variation in disease prevalence among sites and survey times, permutational analysis of variance (PERMANOVA; 9,999 permutations) was conducted in the *R* package *vegan* [43,44], with Bonferroni-corrected pairwise comparisons using the package *pairwiseAdonis* [45]. Non-parametric tests were implemented for all analyses as datasets were non-normal and normal distributions could not be achieved through transformation.

## Fate-tracking of SCTLD-affected *M. cavernosa*

Benthic survey data indicated that disease incidence was too low at sites in Jupiter and Palm Beach, and that coral abundance was too low at sites in St. Lucie Reef, to conduct statistically robust fate-tracking studies in these locations. Consequently, three fate-tracking sites were established in Lauderdale-by-the-Sea (T328, BC1, and FTL4). These three sites are ~12 km from the nearby Hillsboro Inlet, less than 500 m from shore, and have been previously used for benthic and coral monitoring [46–48] within the NFRT (Fig 2). *Montastraea cavernosa* was selected for colony fate-tracking due to the abundance of infected colonies within the study sites. This coral species is considered intermediately susceptible to SCTLD, with onset of tissue loss occurring weeks to months later than what has been reported for highly susceptible species (e.g. *Dendrogyra cylindrus*, *Meandrina meandrites*, *Colpophyllia natans*). Lesions on infected *M. cavernosa* generally progress slower as compared to highly susceptible coral species, with total mortality occurring within months to years [16]. *Montastraea cavernosa* comprised 11.6% of the reported cases of SCTLD in the NFRT in late 2017 [42].

Twenty-four colonies of *M. cavernosa* affected with SCTLD were tagged with uniquely numbered cattle tags across the three sites on 24-August-2018 ($T_1$). Sites were revisited on 11-Sep-2018 ($T_2$), 8-Nov-2018 ($T_3$), and 17-Dec-2018 ($T_4$). Continuous video was taken for 3D model generation to quantify total colony surface area and disease lesion area for each time point.

Fate-tracked colonies were filmed using methods outlined in Young et al. [41], with the following modifications: Canon G16 cameras in Fantasea underwater housings were set on "Underwater mode," 1080p and 60 frames per second (fps), and exposure was adjusted as needed based on ambient light conditions. One-meter, L-shaped PVC scale bars marked at 10 cm increments were placed at opposing right angles to frame the designated colony. A SCUBA diver swam approximately 1 m above the highest point of each coral colony and recorded continuous video while swimming repeated passes in a lawnmower pattern with the camera

pointed directly downward. The number of adjacent passes varied depending on colony size, allowing for 60–70% overlap of filmed surface area. The camera was rotated 90˚ at the end of the first set of adjacent passes and another set of passes was completed perpendicular to the first set. The two complete sets of passes for a single colony required between 1–3 min of dive time, depending on colony size. Each sampling event produced an average of 14 GB of.mp4 video files, or approximately 425 MB of video file size per coral colony.

Video processing and 3D model generation protocols are described in full in our GitHub repository [49]. In summary, the free software package, *FFmpeg* (www.ffmpeg.org), was used to extract still frame images from videos of the fate-tracked colonies at a rate of 3 fps. Still images were then imported into Agisoft Metashape Standard Edition (Version 1.5.2, Agisoft LLC) software, which uses a proprietary algorithm that incorporates SfM and Brown's lens distortion model to generate 3D models from 2D images [31,50]. Model generation in Metashape was conducted according to the manufacturer's protocol in four general steps: 1) camera alignment, 2) dense point cloud generation, 3) mesh generation, and 4) texture overlay. Models were rendered on a 2018 Apple MacBook Pro with a 2.9 GHz processor, 16GB of RAM and a Radeon Pro Vega 16 4GB graphics card. A single model took approximately 40 min to render depending on the number of still images generated. Generated models were then exported as a.obj file and imported into the software Rhinoceros 3D (Robert McNeel & Associates) for analysis; the mean model file size was 64 MB.

Models were scaled using the PVC scale bars, then total colony surface area and disease lesion surface area were measured by hand-tracing polygons around coral tissue. SCTLD disease lesion area was defined as the stark white newly-dead coral tissue or skeleton with sloughing tissue that is characteristic of the disease margin. Both total colony area and disease lesion area were generated directly within Rhinoceros 3D, while healthy tissue area was calculated by subtracting disease lesion area from total colony area. Proportion of loss and rate of tissue loss per week were calculated for each pair of consecutive time points. To identify significant effects of time on healthy area, diseased area, total area, the proportion of tissue loss and the rate of tissue loss, Friedman's rank sum tests were conducted using the package *PMCMRplus* [51]. Pairwise comparisons were made with Nemenyi tests in the package *PMCMR* [52]. Additionally, Spearman's rank correlation analyses were used to correlate diseased area and total colony area and rate of tissue loss and total colony area.

## 3D model validation

Accuracy of model-generated surface area metrics was assessed by a validation experiment using a standardized square cupola prism with equal dimensions and surface features to represent a coral. The prism was constructed from ½" PVC using a top square of 25.4 x 25.4 cm, a bottom square of 47.6 x 47.6 cm, and height of 30.5 cm (S1 Fig). Angled sides were achieved using 45˚ PVC tees to better represent a mounding coral colony. Additionally, polygons of known surface areas (square: 40.32 cm$^2$; rectangle: 12.90 cm$^2$; circle: 45.60 cm$^2$) were printed to scale on waterproof paper, and affixed to the top and sloped sides of the weighted prism.

The prism was filmed in a pool and at the three coral fate-tracking sites in Lauderdale-by-the-Sea (Fig 2) to quantify model error in pristine (pool) versus reef conditions (e.g. variable turbidity, currents, surge). Eight videos were recorded for the prism and boundary frames across the three reef sites (2–3 replicates per site), while six prism videos were captured in the pool. The prism was filmed at a depth of 2.8 m in the pool, 8.8 m at BC1, 5.2 m at T328, and 6.7 m at FTL4; filming distance remained the same for all videos. To assess accuracy of surface area measurements, 3D models were generated and analyzed using the same protocols described earlier, with polygons traced for each of the prism shapes based on the four corners

for squares and rectangles, and the entire circumference for circles. Replicate surface area measurements of shapes were made for each model according to face of the prism (top and four sides), and number of shapes on each face (four for square and rectangle, one for circle), for a total $n = 229$ measurements for square, $n = 232$ for rectangle, and $n = 59$ for circle. Shape error was calculated as the absolute difference between measured shape surface areas and the corresponding template area. Shape error measurements were found to be highly right-skewed, therefore a square-root transformation was applied prior to statistical analyses. Multivariate homogeneity of variance was first tested using the *betadisper* function of the package *vegan*, and following identification of heterogeneous variance among sites due primarily to lower variance at site FTL4. A single-factor PERMANOVA was conducted across sites using the *adonis* function of the package *vegan*.

Additionally, a subset of four models from the colony fate-tracking dataset were extracted at three different frame rates (3–5 fps) across their respective time points to determine if frame rate affected model accuracy. A Kruskal-Wallis test was used to determine if model size differed across the different extraction frequencies.

## Results

### Disease prevalence surveys

SCTLD prevalence varied significantly across location (Table 1), but not through time (Table 1, Fig 3). Pairwise comparisons indicated that disease prevalence was lower in JUP and PB than in SLR and LBTS (Table 1, Fig 3). In LBTS, disease prevalence remained relatively constant (10.9%) while in JUP and PB prevalence was consistently low, but peaked in March 2019 with SCTLD prevalence at 8.1% and 4.3%, respectively. Prevalence data from SLR was highly varied due to generally low coral cover observed during the survey period. For example, the highest reported disease prevalence of 43% at site SLR North in April 2019 was due to low coral cover; only 7 living corals were observed, 3 of which were diseased. The most abundant species throughout our surveys were *M. cavernosa* (63%, $n = 3241$), *Porites astreoides* (17%, $n = 885$), *Siderastrea siderea* (4%, $n = 199$), *Stephanocoenia intersepta* (2.3%, $n = 118$), and *Agaricia agaricites* (2.1%, $n = 108$). Colony abundance by species varied among the four locations (Kruskal-Wallis, $H = 10.308$, $p = 0.016$) with significantly lower abundance between SLR

**Table 1. Results from univariate-permutational analysis of variance (PERMANOVA) comparing SCTLD prevalence from the roving diver surveys across location and time.**

| Test | Comparison | df | Psuedo-*F* | *p*-value[*] |
|------|-----------|-----|-----------|-----------|
| PERMANOVA | Location | 3 | 7.44 | < 0.001 |
| | Date | 12 | 1.23 | ns |
| | Location:Date | 7 | 0.52 | ns |
| Pairwise | LBTS–SLR | | 0.11 | ns |
| | LBTS–JUP | | 26.76 | < 0.001 |
| | LBTS–PB | | 35.91 | < 0.001 |
| | SLR–JUP | | 7.12 | 0.0492 |
| | SLR–PB | | 7.88 | 0.045 |
| | JUP–PB | | 0.63 | ns |

[*]Non-significant *p* values listed as "ns".

Pairwise comparisons from results of the univariate PERMANOVA across all four locations: St. Lucie Reef (SLR), Jupiter (JUP), Palm Beach (PB), and Lauderdale-by-the-Sea (LBTS).

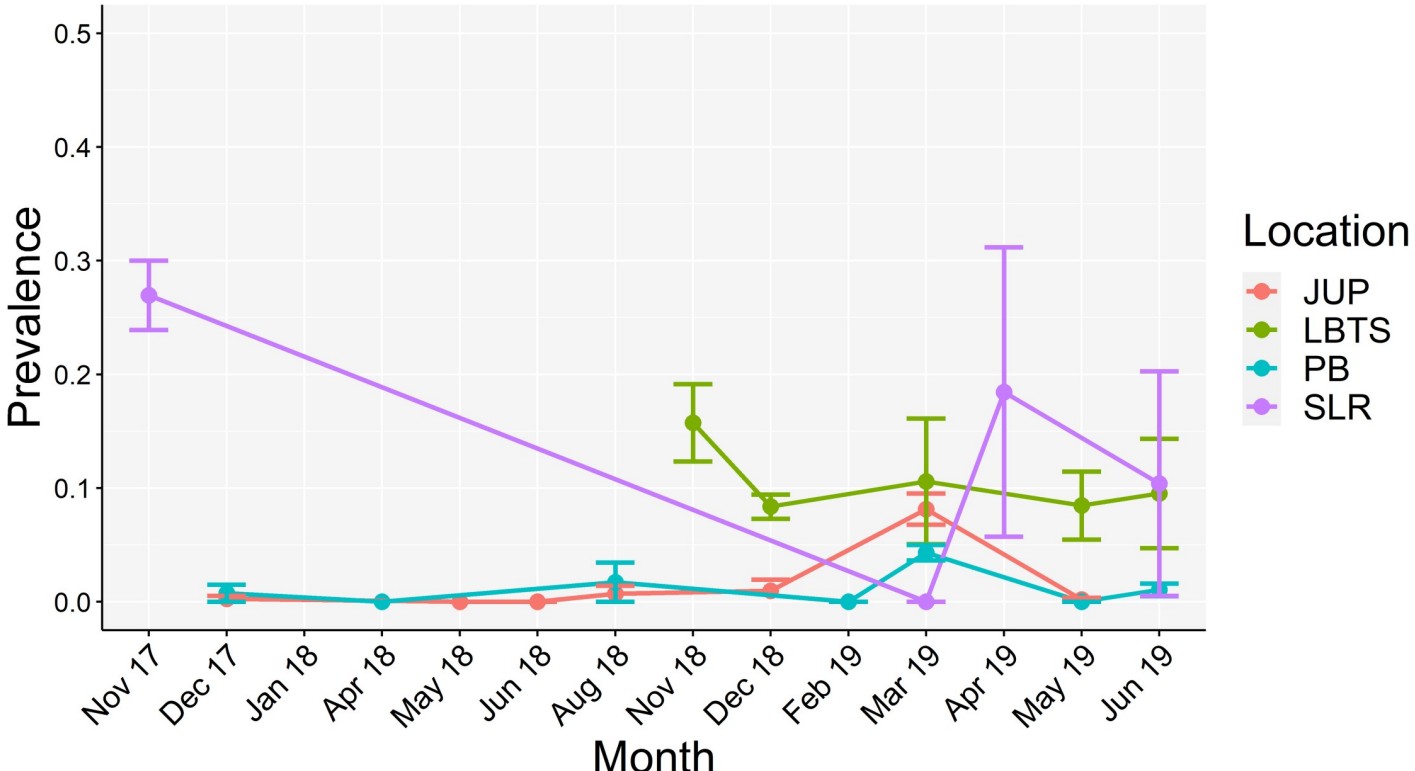

**Fig 3. Mean SCTLD prevalence across all four locations (St. Lucie Reef, Jupiter, Palm Beach and Lauderdale-by-the-Sea).** Points represent means; bars represent standard error.

and all other locations (Dunn's Test, all $p < 0.05$). *Montastraea cavernosa* was the most abundant species at all locations, except for SLR, where *P. astreoides* was the most abundant species (65%, $n = 304$). The species with more than 10 observations that had the highest disease prevalence were *Pseudodiploria clivosa* (36%, $n = 47$), *A. agaricites* (10%, $n = 108$), *Dichocoenia stokesii* (9%, $n = 23$), *Orbicella faveolata* (6%, $n = 49$) and *M. cavernosa* (5%, $n = 3241$).

### Fate-tracking of SCTLD-affected *M. cavernosa*

Friedman's rank sum tests indicated a significant decrease in total colony area (cm$^2$) and healthy tissue area over time (Table 2, Fig 4). Disease lesion area varied significantly through time, however pairwise comparisons revealed that one time point comparison (T$_2$–T$_4$) was driving the variation (Table 2).

A Friedman's rank sum test revealed significant differences between rates of tissue loss through time (df$_{2,71}$, Friedman $\chi^2 = 6.25$, $p = 0.044$). Average tissue loss (mean ± SEM) between T$_1$–T$_2$ was -98.2 ± 40.7 cm$^2$ wk$^{-1}$, -25.0 ± 7.7 cm$^2$ wk$^{-1}$ between T$_2$–T$_3$, and -45.1 ± 19.6 cm$^2$ wk$^{-1}$ between T$_3$–T$_4$ (Fig 5A). Rate of tissue loss was not significantly different among sites between any time point (S1 Table). On average, the proportion of tissue lost was 37.1 ± 7.2% over the course of this study, with three colonies experiencing complete mortality. There was no correlation between total colony area and disease lesion area (Spearman's rank correlation $r_s = 0.15$, $p = 0.154$, Fig 6), or between rates of tissue loss and total colony area (Spearman's rank correlation $r_s = -0.046$, $p = 0.704$, Fig 5B).

**Table 2. Results of Friedman's rank sum tests comparing total colony area, disease tissue area and healthy tissue area across time, with pairwise comparisons made using Nemenyi tests.**

| Test | Comparison | Test Statistic | p value* |
|---|---|---|---|
| Total Colony Area | Friedman's | 48.55 | < 0.001 |
| | $T_1 - T_2$ | 4.27 | ns |
| | $T_1 - T_3$ | 7.59 | < 0.001 |
| | $T_1 - T_4$ | 9.01 | < 0.001 |
| | $T_2 - T_3$ | 3.32 | ns |
| | $T_2 - T_4$ | 4.74 | 0.026 |
| | $T_3 - T_4$ | 1.42 | ns |
| Healthy Tissue Area | Friedman's | 41.29 | < 0.001 |
| | $T_1 - T_2$ | 4.27 | 0.026 |
| | $T_1 - T_3$ | 7.59 | < 0.001 |
| | $T_1 - T_4$ | 9.01 | < 0.001 |
| | $T_2 - T_3$ | 3.32 | ns |
| | $T_2 - T_4$ | 4.74 | ns |
| | $T_3 - T_4$ | 1.42 | ns |
| Disease Lesion Area | Friedman's | 14.02 | 0.003 |
| | $T_1 - T_2$ | 4.27 | ns |
| | $T_1 - T_3$ | 7.59 | ns |
| | $T_1 - T_4$ | 9.01 | ns |
| | $T_2 - T_3$ | 3.32 | ns |
| | $T_2 - T_4$ | 4.74 | 0.012 |
| | $T_3 - T_4$ | 1.42 | ns |

$T_1$, 24-Aug-2018; $T_2$, 11-Sep-2018; $T_3$, 8-Nov-2018; $T_4$, 17-Dec-2018.

*Non-significant p values listed as "ns".

## 3D model validation

Absolute and relative shape error did not vary across the pool and three coral fate-tracking sites in Lauderdale-by-the-Sea (PERMANOVA: $F_{3,279} = 0.733$, $p = 0.584$). The absolute shape error across all sites was $1.7 \pm 0.1$ cm$^2$ (pool: $1.4 \pm 0.1$ cm$^2$; BC1: $2.0 \pm 0.2$ cm$^2$; T328: $2.4 \pm 0.3$ cm$^2$; FTL4: $1.8 \pm 0.2$ cm$^2$), which corresponded to a relative shape error of $6.1 \pm 0.3\%$ (pool: $4.9 \pm 0.3\%$; BC1: $7.0 \pm 0.7\%$; T328: $8.7 \pm 0.9\%$; FTL4: $6.9 \pm 0.6\%$). Average absolute shape error across all sites was $1.70 \pm 0.09$ cm$^2$, which corresponded to relative error of $6.13 \pm 0.27\%$.

To assess potential effects of different frame rates used during model generation on colony-scale surface area measurements, a single-factor Kruskal-Wallis test indicated that frame rate had no significant impact on measured colony surface area (Kruskal-Wallis, $H = 0.12$, $p = 0.94$). This suggests that higher frame rates can be used from the initial video to improve poorly constructed models without affecting downstream spatial analyses. To keep processing time as low as possible, however, stills were extracted at 3 fps unless otherwise necessary.

## Discussion

### Disease prevalence surveys

SCTLD is a unique coral disease due to its broad geographic extent, the number of coral species affected, and rapid rates of disease progression and spread [16,17,21,53]. SCTLD has been reported across ~400 km of the FRT, and now has been observed in at least 12 other territories throughout the TWA [20]. The geographic distribution of SCTLD continues to expand with

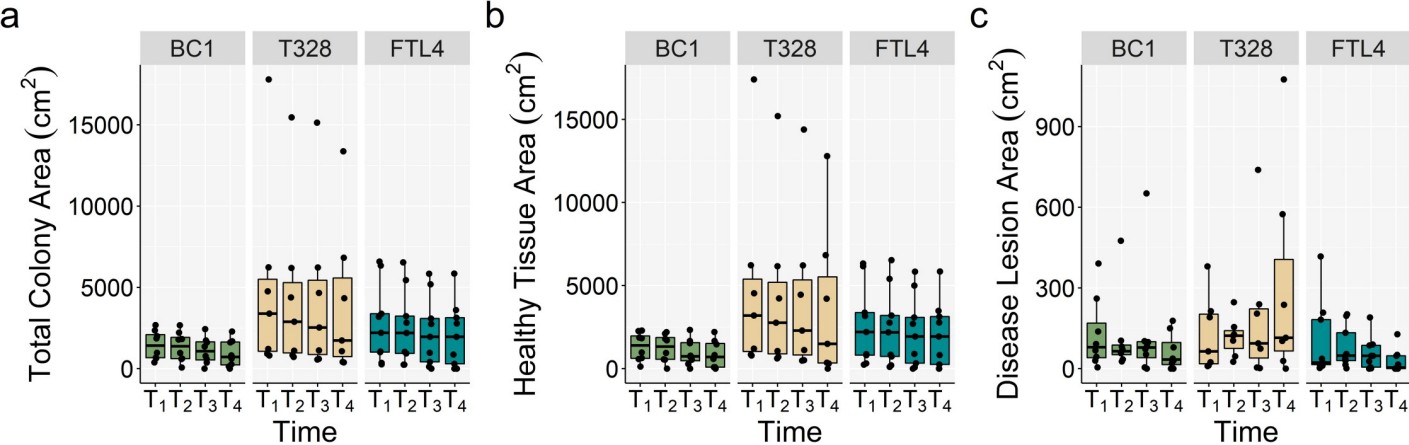

**Fig 4. Comparison of tissue area through time.** Mean tissue areas through time (a) mean total colony area (cm$^2$) (b) mean healthy tissue area (cm$^2$), and (c) mean disease lesion area across all sites and through time for fate-tracked *M. cavernosa* colonies ($n$ = 24). T$_1$, 24-Aug-2018; T$_2$, 11-Sep-2018; T$_3$, 8-Nov-2018; T$_4$, 17-Dec-2018.

**Fig 5. Comparison of rates of tissue loss.** (a) Mean change in tissue area on fate-tracked *M. cavernosa* colonies and (b) Spearman's rank correlation between rate of tissue loss (cm$^2$wk$^{-1}$) and total colony area (cm$^2$).

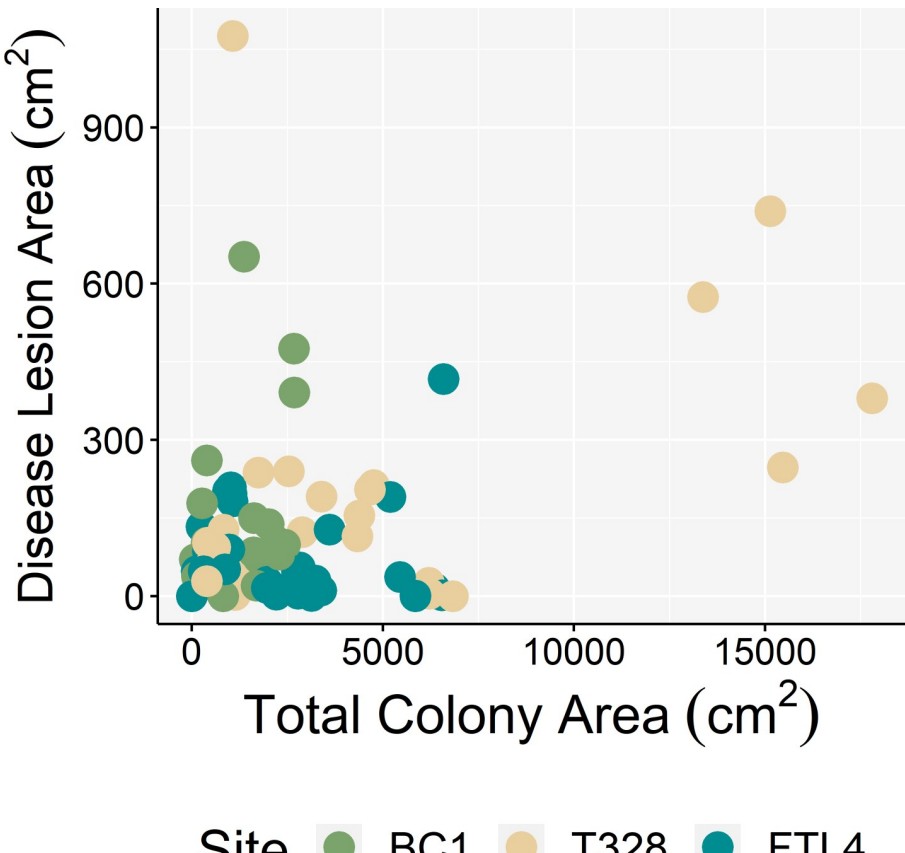

**Fig 6. Correlations between total colony area and disease lesion area.** Spearman's rank correlation between disease lesion area ($cm^2$) and total colony area ($cm^2$) for fate-tracked *M. cavernosa* colonies ($n = 24$).

new observations throughout the Caribbean [19,20]. SCTLD has been reported as far west as Belize and as far southeast as Martinique which are 1300 and 2500 km respectively from the northern terminus of the NFRT at St. Lucie Reef, FL [20]. Other coral diseases, such as white syndrome, have been observed over broad spatial extents (~1500 km) in the Great Barrier Reef [54]. Likewise, black band disease has been observed on coral reefs across ocean basins [55]. The overall mean disease prevalence for the NFRT observed in this study was 6%, which is lower than recorded in post Hurricane Irma surveys in 2017 [42] and previous SCTLD surveys in SE Florida [53]. Sites in Jupiter and Palm Beach had relatively low disease prevalence (1.7% and 1.0%) consistent with background disease levels (1–3%) within the TWA [56]. The highest values observed in this study were between 21–43% at SLR, but no site reached the highest reported disease prevalence values of 60% observed near Miami in 2014 [17]. The lower prevalence values reported in this study may be due in part to differences in species composition between the NFRT and southern regions of the FRT. The most abundant and susceptible species such as *M. meandrites*, *D, stokesii*, and *Pseudodiploria strigosa* [17,53] were comparatively sparse within our NFRT surveys. However, previous SCTLD impacts before our monitoring sites were established may also have affected the relative abundance of susceptible species [53,57].

We observed an increase in disease prevalence during the spring of 2018 which was unexpected, as prevalence for other described coral diseases such as white syndrome, white band, black band, and white pox often increases during the summer months as water temperatures

increase [58–62]. SCTLD prevalence does not appear to have a positive correlation with temperature [53,58] as has been observed for other coral diseases [63–65], but potential environmental cofactors that may drive SCTLD prevalence need to be examined further.

## Fate-tracking of SCTLD-affected *M. cavernosa*

At the colony level, disease progression was highly variable across fate-tracked *M. cavernosa* colonies near Lauderdale-by-the-Sea. As expected, total colony area and healthy tissue area decreased significantly over time, demonstrating the impact SCTLD can have on living coral cover over just a few months. Over the course of this study (115 days), colonies lost on average 37.1 ± 7.24% of their tissue surface area. Using a similar 3D photogrammetry method, Meiling et al. [58] saw proportional losses of 2.0 ± 0.11% day$^{-1}$ of six SCTLD-infected *M. cavernosa* colonies in the US Virgin Islands. Additionally, Aeby et al. [25] noted subacute tissue loss in *M. cavernosa* was 34 ± 8.7% over 1 year using a semi-quantitative method to estimate tissue.

Disease lesion area did not correlate with total colony area, suggesting that larger colonies do not exhibit a higher proportion of diseased tissue compared to smaller colonies. As SCTLD progresses on coral colonies, loss of healthy tissue appears to be a more important indicator of disease virulence, rather than increase in disease lesion area. Importantly, rates of tissue loss did not correlate with total colony area. Therefore, larger corals may have a more favorable time horizon for potential intervention actions that could prevent total colony mortality. Conversely, smaller infected colonies may succumb more quickly to SCTLD. Considering the level of effort required for SCTLD interventions [66], intervention efforts focused on larger colonies with sufficient area of uninfected tissue remaining may result in more successful disease mitigation, and therefore reduction of coral cover loss due to colony mortality.

## 3D model generation as a fate-tracking method

This study tested a relatively inexpensive and rapid 3D model generation methodology described in Young et al. [41] as a means to track disease progression on a colony-level without substantial increases in expense, time, or computational requirements compared to observational and photographic techniques. Healthy and diseased surface area of 24 coral colonies were quantified and compared over time. Additionally, this 3D modeling technique was validated through the quantification of model error using a mock coral colony with standardized surface shapes. Shape error (i.e. variation between measured surface areas and template shapes) did not vary across different depths and turbidity conditions (S2 Fig). This The relative error of 6.13 ± 0.27% is likely an inflated estimate of model error, as the prism shapes used for accuracy assessments were printed on waterproof paper that could move with currents, rather than representing solid surfaces. There were some instances, however, where model generation was poor due to anomalies in the Metashape algorithm, inconsistent underwater filming, or most common, high turbidity, all of which may impact surface area measurements and therefore affect model error. Typically, these issues could be rectified by extracting still images from the video at an increased frame rate to ensure higher overlap among images, or by collecting replicate videos for each target. Notably, extracting at a higher fps improved poor models, but did not disproportionately affect surface area measurements from models that were constructed well using images extracted at 3 fps.

Due to Metashape's proprietary algorithm, further adjustments within the model generation process to rectify poor models are limited. Stable, high-resolution image collection is therefore integral to successful downstream model generation. Alternatives to the lawn-mower video path used in this study may be more effective but also more time-intensive, particularly for tall (>1 m), highly rugose coral colonies. Discrete, overlapping photographs could be taken

instead of continuous video [37], or video could be taken while swimming a circular pattern around the coral at varying angles [67]. Critically, while disease monitoring via linear extension measurements may be able to determine potential differences among colonies or treatments over time [16,28,68], linear extension does not accurately quantify tissue loss and may underestimate the progression of disease lesions on coral colonies. Quantitative 3D approaches such as the method presented here will improve our understanding of the ecology and impacts of SCTLD and other diseases on coral reef ecosystems, and may guide colony selection in future disease intervention strategies.

## Conclusions

SCTLD affects a broad range of host taxa over broad geographic ranges relative to other described coral diseases. Based on the results from this study, larger colonies should be prioritized for SCTLD mitigation measures (i.e. disease intervention approaches) to maximize effort and potential treatment success when logistics and resources may be limited. Since SCTLD is a progressive and necrotic infection, the area of tissue loss, or proportion of tissue loss, may represent more impactful metrics for quantifying the severity of infection as opposed to disease lesion area or percent affected tissue. Additional studies incorporating longer timescales and multiple species are desirable to confirm these patterns of disease progression across all susceptible species. Rapid, cost-effective, and accurate methods using 3D models for quantifying coral surface area are a valuable approach for colony fate-tracking if high-resolution imagery can be obtained. It is recommended that managers and intervention specialists—particularly those focusing on SCTLD—adopt photogrammetric methods to enhance colony tracking methods and facilitate comparability across future studies and intervention trials.

## Supporting information

**S1 Fig. Panel of prism template and deployment for underwater 3D photogrammetry.** A) Template used for five sides of prism, scaled to produce replicates of three standardized shapes: square (40.32 cm$^2$), rectangle (12.90 cm$^2$), and circle (45.60 cm$^2$) simulating surface area measurements on a coral colony. B) Deployment of prism and scaling frames with diver recording continuous video in a lawnmower-pattern at an approximate distance of 1 m. C) Overhead view of prism and scaling frames, with 10 cm banded tape for scaling. D) Deployment of prism and scaling frames on a reef environment at Lauderdale-by-the-Sea, FL.
(TIF)

**S2 Fig. Panel of error measurements for each of the three prism shapes (square, rectangle, circle), calculated as the absolute difference between measured shape surface areas and template areas.**
(TIF)

**S1 Table. Kruskal-Wallis test comparing rate of tissue loss and site.** Non-significant $p$ values are listed as "ns".
(DOCX)

**S1 Dataset. Dataset containing disease prevalence data from roving diver disease surveys.**
(XLSX)

**S2 Dataset. Dataset containing tissue area measurements from rendered 3D models of fate-tracked *Montastraea cavernosa* colonies.**
(XLSX)

**S3 Dataset. Dataset containing shape surface area and error measurements from prism deployment across three sites (pool, Lauderdale-by-the-Sea, St. Lucie Reef).** Error was calculated as the absolute difference between measured surface areas and actual area of the template shapes, then standardized as a percentage relative to template area. (XLSX)

## Acknowledgments

We thank J. Nelson and M. Roy for assistance with boating and diving operations; J. Beal, E. Shilling, and A. Sturm for assistance with data collection. K. Kerrigan, J. Walczak, B. Walker, and K. Neely provided valuable feedback on the study design. This is contribution number 2292 from Harbor Branch Oceanographic Institute at Florida Atlantic University.

## Author Contributions

**Conceptualization:** Joshua D. Voss.

**Data curation:** Ian R. Combs, Michael S. Studivan, Ryan J. Eckert.

**Formal analysis:** Ian R. Combs, Michael S. Studivan.

**Funding acquisition:** Ian R. Combs, Joshua D. Voss.

**Investigation:** Ian R. Combs, Michael S. Studivan, Ryan J. Eckert.

**Methodology:** Ian R. Combs, Michael S. Studivan, Joshua D. Voss.

**Project administration:** Joshua D. Voss.

**Supervision:** Joshua D. Voss.

**Validation:** Ian R. Combs, Michael S. Studivan.

**Visualization:** Ian R. Combs, Michael S. Studivan.

**Writing – original draft:** Ian R. Combs, Michael S. Studivan, Joshua D. Voss.

**Writing – review & editing:** Ian R. Combs, Michael S. Studivan, Ryan J. Eckert, Joshua D. Voss.

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
