## [Decision Letter · Decision Letter 0]

16 Oct 2020

PONE-D-20-25422

Quantifying impacts of stony coral tissue loss disease on corals in Southeast Florida through surveys and 3D photogrammetry

PLOS ONE

Dear Dr. Combs,

Thank you for submitting your manuscript to PLOS ONE. After careful consideration, we feel that it has merit but does not fully meet PLOS ONE’s publication criteria as it currently stands. Therefore, we invite you to submit a revised version of the manuscript that addresses the points raised during the review process.

Reviewer 1 in particular notes that there is a significant literature on photogrammetry techniques on coral reefs, and further attention to this would improve both the introduction and discussion, rather than relying on citations to other taxa. They also require increased clarity in the methods regarding validation of the accuracy of the model, including what site-specific community parameters would impact the 3D model generation. Reviewer 2 also brings up a salient point here, which is that there was no direct comparison to 2D methods, so claims of increased accuracy are not demonstrated. They do recommend that a relatively rapid post-hoc statisitical comparison with 2D screen grabs could be considered, or more cautious language. I would suggest the former to increase the rigour of your study, but either would be sufficient. Reviewer 1 also questions the clarity of statistical analysis, and believes that further investigation into why shape area and error varies with location is either required, or explained in the text.

Both reviewer 1 and I are curious as to why the grouping of disease prevalence is at the county level rather than the location/site scale, particularly since you do not discuss different management, water quality or land use differences between counties (See Fig 3). By working at the location scale, this would allow comparison of more similar replicates, and would also be more accessible to a person not familiar with the geography of Florida. If you feel that county is an important part of the puzzle that makes it a more appropriate unit of replicate than location (more than just indicating a north - south gradient for example), it would be useful to explain this decision. 

Both reviewers identify the recommendation of 'culling' as unjustified, or presented with too high a level of certainty, particularly since the effort, efficacy and description of what other interventions are available are not given (if there is only one colony in an area infected, but it is large, why not remove that if it leads to a better outcome than other interventions?). Without evidence presented here, or citations to small colonies dying more quickly or being disease vectors, this management recommendation goes far beyond the work.  Indeed, in Fig S2 It could be argued that there is some evidence that some larger colonies might lose tissue at a faster rate than some small colonies. In the discussion you also state that "disease lesion area did not correlate with total colony size suggesting that larger colonies do not exhibit a higher proportion of diseased tissue" - this could imply they show the same proportional loss or lower. I strongly suggest you present more data on proportional loss of colony tissue (not just absolute loss) if this is an argument you wish to make. Even so, suggesting culling small colonies as a management measure would still be extreme and would require a stronger argument with references to small colonies as vectors or disease reservoirs. The argument also begs the question - what is a small colony? Nor are what interventions that might be possible shy of culling the entire colony described. 

Finally, there is a significant amount of relevant context and data in the SI. It would be good to see some of the tables and figures into the main text, particularly concerning the disease incidence, colony size and disease spread data.

We look forward to receiving your revised manuscript.

Kind regards,

Fraser Andrew Januchowski-Hartley, Ph.D.

Academic Editor

PLOS ONE

Journal Requirements:

2.  We note that Figure 2 in your submission contain map images which may be copyrighted. All PLOS content is published under the Creative Commons Attribution License (CC BY 4.0), which means that the manuscript, images, and Supporting Information files will be freely available online, and any third party is permitted to access, download, copy, distribute, and use these materials in any way, even commercially, with proper attribution. For these reasons, we cannot publish previously copyrighted maps or satellite images created using proprietary data, such as Google software (Google Maps, Street View, and Earth). For more information, see our copyright guidelines: http://journals.plos.org/plosone/s/licenses-and-copyright.

2.1.    You may seek permission from the original copyright holder of Figure 2 to publish the content specifically under the CC BY 4.0 license. 

2.2.    If you are unable to obtain permission from the original copyright holder to publish these figures under the CC BY 4.0 license or if the copyright holder’s requirements are incompatible with the CC BY 4.0 license, please either i) remove the figure or ii) supply a replacement figure that complies with the CC BY 4.0 license. Please check copyright information on all replacement figures and update the figure caption with source information. If applicable, please specify in the figure caption text when a figure is similar but not identical to the original image and is therefore for illustrative purposes only.

Reviewers' comments:

Reviewer's Responses to Questions

**Comments to the Author**

1. Is the manuscript technically sound, and do the data support the conclusions?

Reviewer #1: Yes

Reviewer #2: Partly

2. Has the statistical analysis been performed appropriately and rigorously? 

Reviewer #1: No

Reviewer #2: Yes

3. Have the authors made all data underlying the findings in their manuscript fully available?

Reviewer #1: Yes

Reviewer #2: Yes

4. Is the manuscript presented in an intelligible fashion and written in standard English?

Reviewer #1: Yes

Reviewer #2: Yes

5. Review Comments to the Author

Reviewer #1: Reviewer comments:

This study monitors SCTL disease prevalence at four locations in the Florida Reef Tract over three months and fate-tracks colony-level impacts using under water photogrammetry and 3D modelling of Montastrea cavernosa colonies at one location.

While the idea, study design and methodology is good, and I think it is great to see more applications of photogrammetry in coral reef research, I have concerns about the statistical evaluation of some results and especially about the model validation. As the fate-tracking is the main focus and the novelty of this work, a clear and detailed evaluation of the used methodology is important and this section should be improved before resubmission (see comments below).

In the introduction I would recommend that the authors give better credit to prior work on photogrammetry in coral reefs. Reading these papers will also help them to improve their model evaluation and compare the accuracy and precision of their methodology to similar applications.

Additional to the comments below I have attached a PDF with annotations, suggesting smaller changes where I think the text could be improved. Most of these suggestions should be easy to implement. If some time is invested on improving the model evaluation I believe this will be a nice contribution to the growing number of photogrammetry applications on coral reefs.

Introduction:

1) from L88: I think applications on cetacean and elasmobranch research are way out of the scope of this work. Better concentrate on referencing all the exciting research on coral reefs.

There is a lot of work that has been done on individual coral colonies to measure surface area, volume, carbon standing stock, coral growth and erosion rates ... I would recommend to read through them if the authors have not come across them before, also to get some inspiration for the section on accuracy and precision of 3D model building. I know all this literature can be overwhelming, but there is some good stuff in there and it will make your paper stronger. E.g.,

Bythell et al., 2001. Three-dimensional morphometric measurements of reef corals using underwater photogrammetry techniques. Coral Reefs

Cocito, Sgorbini, Peirano, & Valle, 2003. 3-D reconstruction of biological objects using underwater video technique and image processing. JEMBE

Courtney, Fisher, Raimondo, Oliver, & Davis, 2007. Estimating 3-dimensional colony surface area of field corals. JEMBE

Burns, Delparte, Gates, & Takabayashi, 2015. Utilizing underwater three-dimensional modeling to enhance ecological and biological studies of coral reefs.

Figueira et al., 2015. Accuracy and Precision of Habitat Structural Complexity Metrics Derived from Underwater Photogrammetry. Remote Sensing

Gutiérrez-Heredia, D'Helft, & Reynaud, 2015. Simple methods for interactive 3D modeling, measurements, and digital databases of coral skeletons. L&O methods

Lavy et al., 2015. A quick, easy and non-intrusive method for underwater volume and surface area evaluation of benthic organisms by 3D computer modelling. MEE

Ferreira et al. 2017. 3D photogrammetry quantifies growth and external erosion of individual coral colonies and skeletons. Sci Rep

Lange & Perry 2020. A quick, easy and non-invasive method to quantify coral growth rates using photogrammetry and 3D model comparisons. MEE

2) Also, in this section, I don’t think you need to cite aerial and scanner methods to prove your point, and the two studies you cite for coral growth did not use photogrammetry at all. Use Ferreira et al. 2017 and Lange & Perry 2020 if you want to keep this sentence. Alternatively you could explain photogrammetry and SfM first and then give examples of studies on reefs. Also see annotations in the PDF.

3) L. 104: optimized from what basis? You have not developed the photogrammetry technique, but a new application for it. Maybe say: “In this study we used photogrammetry techniques as described in [Young et al. ] in order to develop a new application, i.e to track SCTLD disease progression in M. cavernosa colonies.” or something similar?

Methods:

1) see annotations in PDF to improve clarity of the text.

2) L181: state version of Agisoft Metashape that you used as some versions seem to have issues and it will help for repeatability.

3) L. 215: I do not understand why site-specific “community parameters” (what is this anyways?) or even water quality should affect 3D model generation. This would mean that the method is not accurate and should not be used.

Related to this it does not make sense to compare the area of shapes among locations to evaluate the 3D models. So I am wondering why you did not just photograph the mock colony at the sites where you did the fate-tracking in order to calculate the accuracy of your methodology.

Concerning this there is two parameters you want to check:

A) accuracy, meaning how close to reality is your 3D model. Test this by comparing measured shape areas on your mock corals to known areas and calculate the mean error. (This is potentially how you got your 2.17 cm2?) If you notice that the error is bigger at a more turbid reef site, then you could conclude that the visibility affects the accuracy of model building/measurement. But direct comparison of shape areas among sites does not make sense.

B) precision, meaning how good is the reproducibility of your measurements. For this you should measure the surface of the same shape, or better the same coral colony, several times, including all the hand tracing around colonies etc. Then calculate the error, which will show you if you introduce considerable variability using your workflow.

You should calculate the coefficient of variation (error/average) in order to compare your errors to other studies doing surface area measurements. You cannot compare the error from small shapes (2 cm2) directly to the surface area of the colony.

L 240: I do not understand why these correlations were done and what they would tell us about the accuracy of the method.

Results:

1) I am wondering why did you choose to compare counties instead of locations as shown on the map? The latter would have the advantage that replications are more similar.

2) L 255-260: I don't think the figure supports these statements. I would say something along the line of “in Broward, disease prevalence stayed quite constant at around 10%, while in Palm Beach prevalence was usually very low, but peaked in March 2019 when ... of colonies were affected. Disease prevalence at Martin was very variable due to low numbers of live coral colonies...”

3) L 273: Has it been taken into account that colonies were measured repeatedly? Considering that the areas of your colonies are very different you might either have do more fancy statistics using repeated measures GLM/GAM (maybe using colony ID as fixed factor and site and time as explaining variables? sorry, not an expert myself) or you might have to calculate “loss of area” or “loss % of area” in order to make meaningful comparisons over time.

I know you calculated rate of change in healthy and diseased tissue, which I think might make more sense than comparing actual area. In general it is getting a bit confusing looking at so many parameters. I would suggest to think carefully which parameters are most useful in showing what you are interested in and rather use fewer but explain better what they mean.

E.g., I think the most interesting results and the best order in the section L277-309 would be (add numbers and statistics): “The rate of tissue loss did not differ among sites but was variable over time, with highest loss in the first observation interval. The diseased area however did stay constant over time, indicating that the lesion moves with the infected tissue.” Then add the correlations you think are useful. Not all are I think.

4) L287 and 288: S2 Fig should be S3 Fig?

5) L 312-L334: This whole section does not make sense to me. I do not understand why the areas of shapes should be different depending on location. This definitely does not increase trust in the method! It might in part be a relic of your different sample sizes and tests you do. Why do you run ANOVA then Kruskal Wallis and then 3-way PERMANOVA? I fail to understand what any of these significances mean and it is not explained in the results or discussion sections.

I would suggest to think carefully about what parameters actually tell you something about the accuracy/precision of the analysis (see comments above) and rewrite this section completely after the improved analysis.

Discussion:

This section could be improved by being very clear how the results compare to other studies and what can be said about the implications. See annotations in the PDF

1) L353: It is not clear what you are trying to say here. Also please make clear what you mean with L356-357. Next section can be shortened as suggested in PDF.

2) L363: This paragraph seems to be repeating discussions from the previous paragraph. Maybe you could combine those better without repeating yourself?

I think the order of these two paragraphs could be improved. E.g. “In the FRT, disease prevalence is typically higher (...%) than elsewhere in the TWA (~1-3%) [57]. Sites in Palm Beach in the present study showed relatively low background disease (6%), similar to prevalence across the NFRT after Hurricane Irma in Sept 2017 (6%???) (Walker 2018, 53). Highest values observed in this study were 20-45% at sites in Martin, but did not reach levels of up to 60% as reported in [17], likely due to low abundance of susceptible species after ongoing SCTLD impacts [53,58]”

3) L384: I thought Fig S2 showed that it does NOT correlate? Otherwise your next sentence does not make sense

4) I think the suggestion to cull small colonies is a bit drastic and not supported by your study. If there is other research supporting this approach please cite here.

5) L 384-393 could be condensed down to 1-2 sentences saying management should target larger colonies.

6) From L 403: This whole sections would have to be revised after analysis of accuracy/precision.

7) L406: It is not possible to compare the 2.17 cm2 shape error directly to the much larger colonies. Actually, this error seems very high considering that the shape on your mock coral is not very big?! Does that relate to about 10%? Calculate CV and compare to other studies. Also, if you have an error of about 10% you might want to check if the difference of total area/healthy area between time points would still be significant.

8) L409: you did not test the effect of colony size on model accuracy. Also, it does not make sense that water quality (do you mean turbidity/visibility?) affects the size of colonies. You are probably trying to say that lower visibility could result in lower model quality affecting measurements of surface area? Be precise in your wording.

Conclusions and general:

1) Different kinds of interventions should be mentioned in the introduction or discussion if it is the main discussion point in the conclusion.

2) L435: Why would interventions be unsuccessful in small colonies? They should work the same, just preserve less total area, right? So I understand prioritising big ones, but in sites were prevalence is low why not treat small ones too.

3) I think the discussion and conclusion sections will benefit from a second round of review after the improved model evaluation. Try to be very clear what the novelty of this work is (new application of photogrammetry method to accurately quantify tissue loss/disease progression over time) and what can be concluded from it (improve survey protocols, evaluate success of intervention methods... I don’t think culling of half the colonies is a good outcome here).

4) If you want people to take up this method, make the workflow easier to access. I know it is all in the GitHub repository, which is awesome, but I did not easily find the step-by-step guide and what you actually did to measure the areas. Maybe you could prepare a one-pager which is easy to print, stating the steps of image acquisition, model building and analysis to go into Suppl Methods. Or put a link in the methods which leads straight to the guide instead of the whole repository.

5) Also, it would be great if you could make the models (as .obj?) available in the repository.

Reviewer #2: The manuscript by Combs and colleagues presents the results of prevalence assessments and a time-series study of the progression of stony coral tissue loss disease (SCTLD) in the northern Florida Keys. The manuscript also presents the results of an assessment of the accuracy of a 3D photogrammetry modelling technique to track disease lesion progression, and describes the optimised method for use in future surveys. This manuscript was clear, concise, thorough, well organized, and well written. It provides sufficient detail for reproducibility studies and is timely given the on-going outbreak of this virulent and devastating coral disease. I hope that the method presented here can improve and speed-up survey techniques to improve our knowledge and ability to manage the disease.

I don’t have any significant concerns regarding any aspect of the manuscript. The most substantive comment I have is in reference to the conclusion that the model method presented here is “more accurate… than previously established methods”, when there wasn’t actually a quantitative comparison of this method with those previous methods. Thus, I think the conclusion is overreaching (summarised further below). This can either by addressed by a minor edit/softening of the language or a post-hoc statistical comparison of lesion size estimates made from the 2D video screen grabs. Otherwise, I only have minor comments and commend the authors on a well-done study.

General comments

- One of the major findings of this study is that the 3D model method presented is “more accurate data than previously established methods such as two-dimensional surface area estimation (ln 398-399).” I thought the assessment of the accuracy of the 3D method presented here was excellent and rigorous, but there was not a estimate of the disease lesions or the calibration templates/mock coral made from the same two-dimensional photographs to make a robust and unbiased statistical comparison of the two techniques. While I agree that this 3D model pipeline is achievable, it does still require ~40 min per colony of rendering, plus fairly expensive software and hardware for modelling and storage of large video files compared with the much simpler 2D method, so a quantitative comparison of the two methods would strengthen the argument and add justification for using the 3D model method presented here.

- The inclusion of more data from the disease surveys in the main text could be useful, and would balance the disease ecology with the methodological aspects of the study a bit better. Including which species were (and were not) affected at each site, and how disease prevalence changed over time by taxa within each site would be a nice addition to the main text. Line 365-7 mentions the “low abundance and species richness in Marin County sites compared with PB and Broward Counties”, but what is the density and species richness within each site?

- The manuscript is relatively sparse wrt figures and tables, with the vast majority of the information presented as supplementary. I would suggest including a few of these in the main text, particularly Fig S2 and possibly S4, again to balance out the ecology/methodology aspects of the manuscript.

Specific comments

- Line 193 – it is quite tricky to determine whether the stark white area of the lesion is still alive or newly dead, especially in an image/video. Would it be more appropriate to define the lesion area as the ‘stark white coral tissue or very newly dead white skeletal area’? If not, how was this distinction made? It is very difficult to tell in Figure 1, but the white lesion areas look like there is no live tissue to me.

- Line 282 – consider rounding the estimates of # of lesions per colony to the nearst whole number as that makes the most conceptual sense (or at least to the 10th which is probably a more appropriate significant digit)

- Line 360 – consider stating what the most susceptible species are.

- Line 384 – was the correlation between rates of tissue loss and total colony size positive or negative? Please specify.

- Line 388 – typo – suggest ‘greater preservation’

- Line 389 – is there justification for the statement that smaller colonies are more likely to succumb to SCTLD? It may take less time to succumb due to the rate of tissue loss, but is the likelihood of dying from the disease actually higher? Has much recovery been observed? I would suggest editing to “smaller infected colonies may succumb more quickly to SCTLD”

- Line 406 – perhaps you can expand on what you mean by ‘sufficient replication’? Does this mean replicate videos of each colony to render multiple models per colony?

- Line 413 – suggest ‘most commonly’

- Line 434 – suggest ‘regarding’ rather than ‘relative to’

- I wonder if it is worth commenting on how this model approach may work for other coral growth morphologies. Mcavernosa is a fairly simple mounding shape, but encrusting, foliose or branching corals may be much more difficult to accurately model?

- I have strong reservations about the mitigation technique of culling small colonies that is suggested in line 438 (and 389, see comment above). I understand that they may die more quickly than larger colonies and could act as sources of the disease. But the data presented here do not indicate that they are more susceptible (are there size-frequency distributions of healthy vs/ diseased corals in a population?), and the small colonies are often offspring of the few remaining colonies that have survived multiple selection events already and may be the best hope of the recovery of resilient communities, especially if there is any evidence that colonies can recover from this disease. Thus I much prefer the second option of focusing intervention resources on larger colonies and would urge caution when considering culling of smaller colonies.

- Fig 1. Typo in fig legend – add ‘a’ before ‘colony’

6. PLOS authors have the option to publish the peer review history of their article (what does this mean?). If published, this will include your full peer review and any attached files.

Reviewer #1: No

Reviewer #2: No

---

## [Author Response · Author response to Decision Letter 0]

18 Feb 2021

Editor:

Reviewer 1 in particular notes that there is a significant literature on photogrammetry techniques on coral reefs, and further attention to this would improve both the introduction and discussion, rather than relying on citations to other taxa. They also require increased clarity in the methods regarding validation of the accuracy of the model, including what site-specific community parameters would impact the 3D model generation. Reviewer 2 also brings up a salient point here, which is that there was no direct comparison to 2D methods, so claims of increased accuracy are not demonstrated. They do recommend that a relatively rapid post-hoc statistical comparison with 2D screen grabs could be considered, or more cautious language. I would suggest the former to increase the rigor of your study, but either would be sufficient. Reviewer 1 also questions the clarity of statistical analysis, and believes that further investigation into why shape area and error varies with location is either required, or explained in the text.

¬-We took only non-scaled reference photographs and thus cannot conduct a 2D comparison with our 3D data, as such we chose to use more cautious language throughout the manuscript.

Both reviewer 1 and I are curious as to why the grouping of disease prevalence is at the county level rather than the location/site scale, particularly since you do not discuss different management, water quality or land use differences between counties (See Fig 3). By working at the location scale, this would allow comparison of more similar replicates, and would also be more accessible to a person not familiar with the geography of Florida. If you feel that county is an important part of the puzzle that makes it a more appropriate unit of replicate than location (more than just indicating a north - south gradient for example), it would be useful to explain this decision. 

-We agree with the editor and reviewers comments and reanalyzed the survey data to group at the Location level.

Both reviewers identify the recommendation of 'culling' as unjustified, or presented with too high a level of certainty, particularly since the effort, efficacy and description of what other interventions are available are not given (if there is only one colony in an area infected, but it is large, why not remove that if it leads to a better outcome than other interventions?). Without evidence presented here, or citations to small colonies dying more quickly or being disease vectors, this management recommendation goes far beyond the work. Indeed, in Fig S2 It could be argued that there is some evidence that some larger colonies might lose tissue at a faster rate than some small colonies. In the discussion you also state that "disease lesion area did not correlate with total colony size suggesting that larger colonies do not exhibit a higher proportion of diseased tissue" - this could imply they show the same proportional loss or lower. I strongly suggest you present more data on proportional loss of colony tissue (not just absolute loss) if this is an argument you wish to make. Even so, suggesting culling small colonies as a management measure would still be extreme and would require a stronger argument with references to small colonies as vectors or disease reservoirs. The argument also begs the question - what is a small colony? Nor are what interventions that might be possible shy of culling the entire colony described. 

-We removed the suggestion of culling from the manuscript.

Finally, there is a significant amount of relevant context and data in the SI. It would be good to see some of the tables and figures into the main text, particularly concerning the disease incidence, colony size and disease spread data.

-We revised our results section to remove unnecessary metrics and focused on the more compelling outcomes from this study. We also included additional figures focusing on correlation between colony size and disease area as well as rate of tissue loss. 

Reviewer 1:

Reviewer #1: Reviewer comments:

1) from L88: I think applications on cetacean and elasmobranch research are way out of the scope of this work. Better concentrate on referencing all the exciting research on coral reefs. 

- We have removed the unnecessary references to cetacean and elasmobranch research and greatly increased our literature review concerning photogrammetry on corals and coral reefs. We appreciate the suggested journal articles.

2) Also, in this section, I don’t think you need to cite aerial and scanner methods to prove your point, and the two studies you cite for coral growth did not use photogrammetry at all. Use Ferreira et al. 2017 and Lange & Perry 2020 if you want to keep this sentence. Alternatively you could explain photogrammetry and SfM first and then give examples of studies on reefs. Also see annotations in the PDF.

- Reference to aerial and scanner methods have been excluded and the focus was shifted to in-water photogrammetry methods, we also corrected the two citations for Ferreira et al. 2017 and Lange & Perry 2020. 

3) L. 104: optimized from what basis? You have not developed the photogrammetry technique, but a new application for it. Maybe say: “In this study we used photogrammetry techniques as described in [Young et al. ] in order to develop a new application, i.e to track SCTLD disease progression in M. cavernosa colonies.” or something similar?

-We revised the wording of this to better represent our work (L109).

Methods:

1) see annotations in PDF to improve clarity of the text.

-The PDF annotations were applied, they were very helpful and the authors thank the reviewer for their efforts to improve the clarity and writing throughout this manuscript. 

2) L181: state version of Agisoft Metashape that you used as some versions seem to have issues and it will help for repeatability.

-This has been included (L181).

3) L. 215: I do not understand why site-specific “community parameters” (what is this anyways?) or even water quality should affect 3D model generation. This would mean that the method is not accurate and should not be used.

- Our intention was to compare surface area error from models constructed in varying environments (pristine pool versus nearshore coral reef with low visibility and high wave action). To make the model accuracy validation section more clear and applicable to the fate-tracking results, we have focused on comparisons of shape error between the pool models and models made at the fate-tracking sites in Lauderdale-by-the-Sea. We have rephrased this section to be more explicit starting at L214. 

Related to this it does not make sense to compare the area of shapes among locations to evaluate the 3D models. So I am wondering why you did not just photograph the mock colony at the sites where you did the fate-tracking in order to calculate the accuracy of your methodology.

- Models with the mock coral colony were constructed at the same fate-tracking sites in the original version of the manuscript, but were not represented as separate sites in the original analysis. To improve the comparability of the model accuracy analyses to the fate-tracking portion, we have focused on the fate-tracking sites in Lauderdale-by-the-Sea only. Likewise, we have removed the analyses of overall shape areas and focused on the shape error associated with each model.

Concerning this there is two parameters you want to check:

A) accuracy, meaning how close to reality is your 3D model. Test this by comparing measured shape areas on your mock corals to known areas and calculate the mean error. (This is potentially how you got your 2.17 cm2?) If you notice that the error is bigger at a more turbid reef site, then you could conclude that the visibility affects the accuracy of model building/measurement. But direct comparison of shape areas among sites does not make sense.

B) precision, meaning how good is the reproducibility of your measurements. For this you should measure the surface of the same shape, or better the same coral colony, several times, including all the hand tracing around colonies etc. Then calculate the error, which will show you if you introduce considerable variability using your workflow.

You should calculate the coefficient of variation (error/average) in order to compare your errors to other studies doing surface area measurements. You cannot compare the error from small shapes (2 cm2) directly to the surface area of the colony.

- We have made substantial efforts in the data analysis and writing to make this section of the manuscript clearer and more applicable to the coral fate-tracking portion. First, we focused on comparison of model error across the pool and the three coral fate-tracking sites in Lauderdale-by-the-Sea. The statistical analyses were revised to a comparison of shape error (i.e. the difference between 3D modeled shape areas and the template) among sites (L214-222, L225-230, L308-3111). Replicate measurements of all the prism shapes were made within the original analyses, but this was not clear in the original version. The prism had five faces with the template shapes (top and four sides), each of which had multiple shapes (four squares and rectangles, one circle per face) (L222-225). Shape error is also represented in two ways, as 1) the absolute difference between measured shape areas and template area, and 2) percent error relative to template area (similar to previous studies, including those you suggested in the original review). The former is used for the statistical tests and for visual assessment of shape error in S4 Fig, and the latter is reported in the text (L311-314). These calculations result in mean errors of 1.70 sq. cm and 6.13%, respectively. The coefficient of variation was not calculated for these analyses, as this is typically used for comparisons of replicate 3D models or measurements. Each of the measurements in our prism dataset are statistically independent, and we feel provide for a robust assessment of model accuracy.

L 240: I do not understand why these correlations were done and what they would tell us about the accuracy of the method.

- This analysis has been removed, and the section has been rewritten according to the revised analysis of accuracy across sites.

Results:

1) I am wondering why did you choose to compare counties instead of locations as shown on the map? The latter would have the advantage that replications are more similar.

-The disease prevalence survey analyses were shifted from county level to location level as shown on the map. 

2) L 255-260: I don't think the figure supports these statements. I would say something along the line of “in Broward, disease prevalence stayed quite constant at around 10%, while in Palm Beach prevalence was usually very low, but peaked in March 2019 when ... of colonies were affected. Disease prevalence at Martin was very variable due to low numbers of live coral colonies...”

-This section has been edited to improve clarity and better reflect our results and figures (L238-255). 

3) L 273: Has it been taken into account that colonies were measured repeatedly? Considering that the areas of your colonies are very different you might either have do more fancy statistics using repeated measures GLM/GAM (maybe using colony ID as fixed factor and site and time as explaining variables? sorry, not an expert myself) or you might have to calculate “loss of area” or “loss % of area” in order to make meaningful comparisons over time.

I know you calculated rate of change in healthy and diseased tissue, which I think might make more sense than comparing actual area. In general it is getting a bit confusing looking at so many parameters. I would suggest to think carefully which parameters are most useful in showing what you are interested in and rather use fewer but explain better what they mean.

E.g., I think the most interesting results and the best order in the section L277-309 would be (add numbers and statistics): “The rate of tissue loss did not differ among sites but was variable over time, with highest loss in the first observation interval. The diseased area however did stay constant over time, indicating that the lesion moves with the infected tissue.” Then add the correlations you think are useful. Not all are I think.

-The Friedman’s test and subsequent post hoc analysis, the Nemenyi test, are appropriate since our data is not normally distributed; it is a non-parametric equivalent to a repeated measures analysis of variance. The section has been reduced in the number of metrics and tests run to increase overall clarity and focus on important, compelling results. (L289-297). Proportion of loss data was added, and the only correlation that was retained, as the authors thought this was the most compelling, was the correlation between disease lesion area and total colony size. 

4) L287 and 288: S2 Fig should be S3 Fig?

- These were removed.

5) L 312-L334: This whole section does not make sense to me. I do not understand why the areas of shapes should be different depending on location. This definitely does not increase trust in the method! It might in part be a relic of your different sample sizes and tests you do. Why do you run ANOVA then Kruskal Wallis and then 3-way PERMANOVA? I fail to understand what any of these significances mean and it is not explained in the results or discussion sections.

- In the revised version of the manuscript, we have streamlined our assessments of model accuracy according to previous suggestions. We hope that the new sections present a better description of the analyses that were conducted. Also, following careful review of the statistical tests and 3D models, we identified a few outliers in the prism dataset that were the result of holes in the mesh leading to inaccurate surface area measurements, which appeared to be driving the significant differences presented in the original manuscript.

I would suggest to think carefully about what parameters actually tell you something about the accuracy/precision of the analysis (see comments above) and rewrite this section completely after the improved analysis.

- Thank you for your constructive comments, we hope that the revised sections address your concerns and suggestions.

Discussion:

This section could be improved by being very clear how the results compare to other studies and what can be said about the implications. See annotations in the PDF

1) L353: It is not clear what you are trying to say here. Also please make clear what you mean with L356-357. Next section can be shortened as suggested in PDF.

- This section, along with the section below, were combined and rewritten to improve clarity (L324-350).

2) L363: This paragraph seems to be repeating discussions from the previous paragraph. Maybe you could combine those better without repeating yourself?

I think the order of these two paragraphs could be improved. E.g. “In the FRT, disease prevalence is typically higher (...%) than elsewhere in the TWA (~1-3%) [57]. Sites in Palm Beach in the present study showed relatively low background disease (6%), similar to prevalence across the NFRT after Hurricane Irma in Sept 2017 (6%???) (Walker 2018, 53). Highest values observed in this study were 20-45% at sites in Martin, but did not reach levels of up to 60% as reported in [17], likely due to low abundance of susceptible species after ongoing SCTLD impacts [53,58]”

-The following section was changed using your suggestions to increase clarity (L332-338).

3) L384: I thought Fig S2 showed that it does NOT correlate? Otherwise your next sentence does not make sense

- This line was edited to correct the typo, the sentence should have read “rates of tissue loss did not correlate” as stated in the reviewer comment. 

4) I think the suggestion to cull small colonies is a bit drastic and not supported by your study. If there is other research supporting this approach please cite here.

- The suggestion of culling has been removed from the manuscript

5) L384-393 could be condensed down to 1-2 sentences saying management should target larger colonies.

-This section was revised to omit culling and express the need to target larger colonies.

6) From L 403: This whole sections would have to be revised after analysis of accuracy/precision.

- Following the revised analysis, we have rewritten this section to emphasize the error rate in the prism measurements (L379-382), the potential causes (flexible shape templates and water movement/poor visibility; L382), and suggestions for improving model construction and underwater filming/photography (L384-391).

7) L406: It is not possible to compare the 2.17 cm2 shape error directly to the much larger colonies. Actually, this error seems very high considering that the shape on your mock coral is not very big?! Does that relate to about 10%? Calculate CV and compare to other studies. Also, if you have an error of about 10% you might want to check if the difference of total area/healthy area between time points would still be significant.

- Based on previous studies, we have opted to calculate error rate (%) for shape measurements relative to the corresponding template area. As a result, the global error rate is ~6%.

8) L409: you did not test the effect of colony size on model accuracy. Also, it does not make sense that water quality (do you mean turbidity/visibility?) affects the size of colonies. You are probably trying to say that lower visibility could result in lower model quality affecting measurements of surface area? Be precise in your wording.

- We have revised this section to be more specific regarding surface area measurements and potential effects of low visibility on model construction and error (L389).

Conclusions and general:

1) Different kinds of interventions should be mentioned in the introduction or discussion if it is the main discussion point in the conclusion.

-There has been a wide variety of intervention methods proposed, and refinement and evaluation is still ongoing, so we did not go into detail on the specific methods, but did include them in the introduction (L77-82). 

2) L435: Why would interventions be unsuccessful in small colonies? They should work the same, just preserve less total area, right? So I understand prioritizing big ones, but in sites were prevalence is low why not treat small ones too.

-The reference to interventions being unsuccessful was removed and clarification was added to imply that larger colonies should be prioritized if present, but smaller colonies should not be ignored all together. 

3) I think the discussion and conclusion sections will benefit from a second round of review after the improved model evaluation. Try to be very clear what the novelty of this work is (new application of photogrammetry method to accurately quantify tissue loss/disease progression over time) and what can be concluded from it (improve survey protocols, evaluate success of intervention methods... I don’t think culling of half the colonies is a good outcome here).

-The discussion and conclusions have been revised to focus more on the novelty of this work, comparable photogrammetry and SCTLD studies, and improving the management and intervention suggestions throughout. 

4) If you want people to take up this method, make the workflow easier to access. I know it is all in the GitHub repository, which is awesome, but I did not easily find the step-by-step guide and what you actually did to measure the areas. Maybe you could prepare a one-pager which is easy to print, stating the steps of image acquisition, model building and analysis to go into Suppl Methods. Or put a link in the methods which leads straight to the guide instead of the whole repository.

- A reference to the protocol within our GitHub repository was added to the methods section. (L178) 

5) Also, it would be great if you could make the models (as .obj?) available in the repository.

-Unfortunately, the models are too large to be housed on the GitHub repository, they will be available upon request. 

Reviewer 2:

General comments

- One of the major findings of this study is that the 3D model method presented is “more accurate data than previously established methods such as two-dimensional surface area estimation (ln 398-399).” I thought the assessment of the accuracy of the 3D method presented here was excellent and rigorous, but there was not a estimate of the disease lesions or the calibration templates/mock coral made from the same two-dimensional photographs to make a robust and unbiased statistical comparison of the two techniques. While I agree that this 3D model pipeline is achievable, it does still require ~40 min per colony of rendering, plus fairly expensive software and hardware for modelling and storage of large video files compared with the much simpler 2D method, so a quantitative comparison of the two methods would strengthen the argument and add justification for using the 3D model method presented here.

-We took 2D reference photographs in order to double check possible abnormalities in the model (eg: if that is a Christmas tree worm or a small disease lesion) they are not scaled and therefore cannot be analyzed and used in a comparison of the two techniques. However, we have incorporated references to previous studies that have compared 3D photogrammetry to 2D and physical methods (tin foil method) that have shown 3D photogrammetry to be a more robust method.

- The inclusion of more data from the disease surveys in the main text could be useful, and would balance the disease ecology with the methodological aspects of the study a bit better. Including which species were (and were not) affected at each site, and how disease prevalence changed over time by taxa within each site would be a nice addition to the main text. Line 365-7 mentions the “low abundance and species richness in Marin County sites compared with PB and Broward Counties”, but what is the density and species richness within each site?

-We feel that adding broader ecological surveys of these reefs outside of SCTLD prevalence is outside the scope of this study. We included more species abundance data, mainly to highlight the lack of ‘highly susceptible’ species within our surveys is possibly due to SCTLD already and how different the coral communities are at SLR relative to the rest of our sites. 

- The manuscript is relatively sparse wrt figures and tables, with the vast majority of the information presented as supplementary. I would suggest including a few of these in the main text, particularly Fig S2 and possibly S4, again to balance out the ecology/methodology aspects of the manuscript.

We included additional figures focusing on correlation between colony size and disease area as well as rate of tissue loss. 

Several Specific comments

- Line 193 – it is quite tricky to determine whether the stark white area of the lesion is still alive or newly dead, especially in an image/video. Would it be more appropriate to define the lesion area as the ‘stark white coral tissue or very newly dead white skeletal area’? If not, how was this distinction made? It is very difficult to tell in Figure 1, but the white lesion areas look like there is no live tissue to me.

- We revised this to improve clarity (L193)

- Line 282 – consider rounding the estimates of # of lesions per colony to the nearst whole number as that makes the most conceptual sense (or at least to the 10th which is probably a more appropriate significant digit)

-These were removed from the analysis .

- Line 360 – consider stating what the most susceptible species are.

-This has been included (L340)

- Line 384 – was the correlation between rates of tissue loss and total colony size positive or negative? Please specify.

-This was a typo, it should have read “did not” correlate, it has been adjusted (L361)

- Line 388 – typo – suggest ‘greater preservation’

- This section has been rewritten

- Line 389 – is there justification for the statement that smaller colonies are more likely to succumb to SCTLD? It may take less time to succumb due to the rate of tissue loss, but is the likelihood of dying from the disease actually higher? Has much recovery been observed? I would suggest editing to “smaller infected colonies may succumb more quickly to SCTLD”

-This section was heavily revised as suggested and the statement regarding small colonies has been omitted.

- Line 406 – perhaps you can expand on what you mean by ‘sufficient replication’? Does this mean replicate videos of each colony to render multiple models per colony?

-This section has been heavily revised and this line has been removed.

- Line 413 – suggest ‘most commonly’

-This section has been heavily revised and this line has been removed.

- Line 434 – suggest ‘regarding’ rather than ‘relative to’

-This has been changed as suggested (L406)

- I wonder if it is worth commenting on how this model approach may work for other coral growth morphologies. Mcavernosa is a fairly simple mounding shape, but encrusting, foliose or branching corals may be much more difficult to accurately model?

- I have strong reservations about the mitigation technique of culling small colonies that is suggested in line 438 (and 389, see comment above). I understand that they may die more quickly than larger colonies and could act as sources of the disease. But the data presented here do not indicate that they are more susceptible (are there size-frequency distributions of healthy vs/ diseased corals in a population?), and the small colonies are often offspring of the few remaining colonies that have survived multiple selection events already and may be the best hope of the recovery of resilient communities, especially if there is any evidence that colonies can recover from this disease. Thus I much prefer the second option of focusing intervention resources on larger colonies and would urge caution when considering culling of smaller colonies.

-The suggestion of culling has been removed throughout the manuscript.

- Fig 1. Typo in fig legend – add ‘a’ before ‘colony’

-This has been changed as suggested.

---

## [Decision Letter · Decision Letter 1]

31 Mar 2021

PONE-D-20-25422R1

Quantifying impacts of stony coral tissue loss disease on corals in Southeast Florida through surveys and 3D photogrammetry

PLOS ONE

Dear Dr. Combs,

Thank you for submitting your manuscript to PLOS ONE. After careful consideration, we feel that it has merit and is very close to appropriate standard for publication. Both reviewers are happy with how you have addressed their critiques and comments, but both do point out some minor errors and comments. In reviewer 1's case in particular, many of these are minor typographically errors or slight clarifications, but due to the process at PLoS One, you would not have the opportunity to correct them at a 'proofing' stage. Figure 5a also does not correspond to the Figure legend, and the panel appears to be a duplicate of Figure 6. Reviewer 2 would also like you to consider some additional disease ecology factors, and please see their attached pdf with some comments.

We look forward to receiving your revised manuscript.

Kind regards,

Fraser Andrew Januchowski-Hartley, Ph.D.

Academic Editor

PLOS ONE

Journal Requirements:

Reviewers' comments:

Reviewer's Responses to Questions

**Comments to the Author**

1. If the authors have adequately addressed your comments raised in a previous round of review and you feel that this manuscript is now acceptable for publication, you may indicate that here to bypass the “Comments to the Author” section, enter your conflict of interest statement in the “Confidential to Editor” section, and submit your "Accept" recommendation.

Reviewer #1: All comments have been addressed

Reviewer #2: (No Response)

2. Is the manuscript technically sound, and do the data support the conclusions?

Reviewer #1: Yes

Reviewer #2: Partly

3. Has the statistical analysis been performed appropriately and rigorously? 

Reviewer #1: Yes

Reviewer #2: Yes

4. Have the authors made all data underlying the findings in their manuscript fully available?

Reviewer #1: Yes

Reviewer #2: Yes

5. Is the manuscript presented in an intelligible fashion and written in standard English?

Reviewer #1: Yes

Reviewer #2: Yes

6. Review Comments to the Author

Reviewer #1: Reviewer 1 Round 2

The authors did a good job addressing my questions and suggestions in their revised version of the manuscript and improved the photogrammetry method section and discussion section significantly. Well done!

Please find below a couple of very minor suggestions I noted down while reading over the manuscript, otherwise I am happy to recommend publication as is.

Check Figures 5 and 6 and the captions and references in the text. Figure 5 includes the disease lesion area plot (but note different font size) but this is not represented in the figure caption and text references and Figure 6 depicts the same plot.

L173: hovered approximately 1 m above

L176: overlap of filmed surface area

L183: (link does not work)

L249: site SLR North in April 2019

L253: What do you mean with species abundance? Overall colony abundance? or number of species? Also, was this parameter higher or lower at SLR (says varying in L254)

L289: tissue area (get rid of s)

L313: This first sentence seems out of place. Maybe better integrates in method section as the reason to use PERMANOVA. Then start with “Absolute and relative shape error did not vary across the pool and ...”

L316: Absolute shape error

L318: corresponded to a relative shape error of

L322: on measured colony surface area

L332-334: consider to move this sentence behind L337 starting with “Other ... observed over similar spatial scales” (as you note similar distance to Belize/Martinique)

L342: no site (get rid of s)

L343: near Miami in ...(year)

L346: comparatively sparse (or absent)

L360: on a coral colony level over just ... over the course of this study

L361: why “in contrast”?

L374: sufficient area (get rid of s)

L379: progression on a colony-level

381: I would not consider this a complex morphology. Would rephrase to “Healthy and diseased surface area of 24 coral colonies were quantified and compared over time”

385: did not vary across different depth and turbidity conditions (S2 Fig.)

L386: Average absolute shape error

L387: to a relative error of ...

L389: rather than representing solid surfaces.

L404: may be able to determine

L406: the progression of disease

L411: what does non-discriminant mean? Do you mean “Compared to other coral diseases, impacts of SCTLD are not yet well described concerning host-specificity and spatio-temporal distribution”? Is that true? Lots of studies out on SCTLD

L412-416: would suggest to shorten and rephrase, e.g.: “Based on the results from this study, larger colonies should be prioritized for SCTLD mitigation measures.”

L417: remove alternative

L419-420:” ... longer timescales and multiple species are desirable to confirm observed patterns ... “ (3D mentioned in next sentence)

L422: not sure what redundancies and precautions you refer to. I suggest “... are a valuable approach for colony fate-tracking if high resolution imagery can be obtained.“

L425: more accurate than what?

Reviewer #2: The manuscript by Combs et al. reports the results of spatial and temporal field surveys of SCTLD in Florida and, using the survey results to inform site selection, undertook fate-tracking of individual Montastrea cavernosa colonies at the most impacted sites. The fate-tracking applied a structure-from-motion 3D photogrammetry technique to quantify rates of tissue loss and changes to lesion area and lesion number through time. The authors then used those data to improve our understanding of disease dynamics and make management recommendations.

The revised manuscript has addressed all the major concerns I raised in my previous review, including removing mention of culling, removing statements of direct comparisons with 2D methods that were not justified by the data, reanalysing the data at the location level, and including some of the supplementary data in the main text. Overall, I think the manuscript has been clarified and streamlined will be a useful addition to the literature on this important topic. I only have a few outstanding comments that warrant a minor review, most notably about the intervention recommendation, and have added some edits and comments in an annotated PDF, which I hope the authors will find useful.

• References: Some key references on disease ecology and SCTLD are missing, including Muller et al. 2020 Spatial Epidemiology of the Stony-Coral-Tissue-Loss Disease in Florida. Front Mar Sci 7:163; Meiling et al. (2020) and stony coral tissue loss disease (SCTLD) lesion progression slows in association with thermal stress. Frontiers in Marine Science, 7, 1128. More references are added in the annotated PDF file, and I urge the authors to review some of the key disease literature for the Caribbean in the discussion of ecological drivers including temperature, local pressures, etc.

• Intervention recommendation: Line 36, Line 374, Line 415: I think it is worth adding a caveat/qualifier here about what the end goal of the intervention is. If the goal is to minimise immediate loss of coral cover, then yes, perhaps it is best to prioritise the largest colonies. But, that doesn’t take into account other on-going processes. Firstly, one could argue that the smaller colonies represent younger and more locally adapted/stress tolerant genotypes, and applying intervention efforts across all colonies in a small high-value (heat tolerant)/well connected (larval source)/high-flow (i.e. pathogen source) area might be better than selecting only larger individuals in a broader area using the same amount of time/resources. Secondly, the disease appears to follow contagious transmission dynamics (Muller et al 2020), so if infected small colonies are left untreated in an area around larger colonies that are treated, it may remain a pathogen source to re-infect the larger colonies; I understand this argument led to your original suggestion of culling but that also doesn’t factor in other processes. I would urge you to add a qualifier as such and soften the language because the statement “should be prioritised” is quite strong when these other factors aren’t considered.

• Methods: Lines 142-147: The sentences “Statistical tests were run in the R statistical environment” and “non-parametric tests were implemented for all analyses unless otherwise noted”… seemed out of place and premature, because it isn’t clear what tests were run and why? I suggest moving line 146 up to start the section with “to assess variation in disease prevalence among sites and survey times”, and then you can explain the methodological details.

• Figures: It appears that Figures 5a and 6 are the same? Also, Figure 5b could be faceted by site to support the argument that tissue loss didn’t differ among sites.

7. PLOS authors have the option to publish the peer review history of their article (what does this mean?). If published, this will include your full peer review and any attached files.

Reviewer #1: **Yes: **Ines Lange

Reviewer #2: No

---

## [Author Response · Author response to Decision Letter 1]

4 May 2021

Response to reviewers: 

Reviewer #1: Reviewer 1 Round 2

The authors did a good job addressing my questions and suggestions in their revised version of the manuscript and improved the photogrammetry method section and discussion section significantly. Well done!

Please find below a couple of very minor suggestions I noted down while reading over the manuscript, otherwise I am happy to recommend publication as is.

- Thank you very much for taking the time to review this manuscript again, and for providing additional edits to improve readability. We have addressed your comments in the line-by-line responses below.

Check Figures 5 and 6 and the captions and references in the text. Figure 5 includes the disease lesion area plot (but note different font size) but this is not represented in the figure caption and text references and Figure 6 depicts the same plot.

- Thank you for catching this, this was the result of a small error in the R code used to generate figures. The correct figures have now been included.

L173: hovered approximately 1 m above

- Changed as suggested

L176: overlap of filmed surface area

- Changed as suggested

L183: (link does not work)

- Link has been fixed

L249: site SLR North in April 2019

- Changed as suggested

L253: What do you mean with species abundance? Overall colony abundance? or number of species? Also, was this parameter higher or lower at SLR (says varying in L254)

- Reworded to signify as colony abundance and lower abundance at SLR relative to all other sites.

L289: tissue area (get rid of s)

- Changed as suggested

L313: This first sentence seems out of place. Maybe better integrates in method section as the reason to use PERMANOVA. Then start with “Absolute and relative shape error did not vary across the pool and ...”

- Changed as suggested

L316: Absolute shape error

- Changed as suggested

L318: corresponded to a relative shape error of

- Changed as suggested

L322: on measured colony surface area

- Changed as suggested

L332-334: consider to move this sentence behind L337 starting with “Other ... observed over similar spatial scales” (as you note similar distance to Belize/Martinique)

- Changed as suggested

L342: no site (get rid of s)

- Changed as suggested

L343: near Miami in ...(year)

- Changed as suggested

L346: comparatively sparse (or absent)

- Changed as suggested

L360: on a coral colony level over just ... over the course of this study

- Changed as suggested

L361: why “in contrast”?

- This has been removed to avoid confusion.

L374: sufficient area (get rid of s)

- Changed as suggested

L379: progression on a colony-level

- Changed as suggested

381: I would not consider this a complex morphology. Would rephrase to “Healthy and diseased surface area of 24 coral colonies were quantified and compared over time”

- Changed as suggested

385: did not vary across different depth and turbidity conditions (S2 Fig.)

- Changed as suggested

L386: Average absolute shape error

- Changed as suggested

L387: to a relative error of ... 

- Changed as suggested

L389: rather than representing solid surfaces.

- Changed as suggested

L404: may be able to determine

- Changed as suggested

L406: the progression of disease

- Changed as suggested

L411: what does non-discriminant mean? Do you mean “Compared to other coral diseases, impacts of SCTLD are not yet well described concerning host-specificity and spatio-temporal distribution”? Is that true? Lots of studies out on SCTLD

- This statement has been rewritten to emphasize that SCTLD has broad taxonomic and temporal impacts relative to other coral diseases.

L412-416: would suggest to shorten and rephrase, e.g.: “Based on the results from this study, larger colonies should be prioritized for SCTLD mitigation measures.”

- Changed as suggested

L417: remove alternative

- Changed as suggested

L419-420:” ... longer timescales and multiple species are desirable to confirm observed patterns ... “ (3D mentioned in next sentence)

- Changed as suggested

L422: not sure what redundancies and precautions you refer to. I suggest “... are a valuable approach for colony fate-tracking if high resolution imagery can be obtained.“

- Changed as suggested

L425: more accurate than what?

- Added language to clarify

Reviewer #2: The manuscript by Combs et al. reports the results of spatial and temporal field surveys of SCTLD in Florida and, using the survey results to inform site selection, undertook fate-tracking of individual Montastrea cavernosa colonies at the most impacted sites. The fate-tracking applied a structure-from-motion 3D photogrammetry technique to quantify rates of tissue loss and changes to lesion area and lesion number through time. The authors then used those data to improve our understanding of disease dynamics and make management recommendations.

The revised manuscript has addressed all the major concerns I raised in my previous review, including removing mention of culling, removing statements of direct comparisons with 2D methods that were not justified by the data, reanalysing the data at the location level, and including some of the supplementary data in the main text. Overall, I think the manuscript has been clarified and streamlined will be a useful addition to the literature on this important topic. I only have a few outstanding comments that warrant a minor review, most notably about the intervention recommendation, and have added some edits and comments in an annotated PDF, which I hope the authors will find useful.

- Thank you very much for taking the time to review this manuscript again, and for providing additional edits to improve readability. We have addressed your comments in the line-by-line responses below.

• References: Some key references on disease ecology and SCTLD are missing, including Muller et al. 2020 Spatial Epidemiology of the Stony-Coral-Tissue-Loss Disease in Florida. Front Mar Sci 7:163; Meiling et al. (2020) and stony coral tissue loss disease (SCTLD) lesion progression slows in association with thermal stress. Frontiers in Marine Science, 7, 1128. More references are added in the annotated PDF file, and I urge the authors to review some of the key disease literature for the Caribbean in the discussion of ecological drivers including temperature, local pressures, etc.

- Thank you for providing additional references to strengthen our conclusions. The Meiling et al. (2020) study was referenced in the first revision, but a glitch with our reference manager software caused a replacement of the Muller et al. (2020) study. We have added it back in, and have included your additional suggestions.

• Intervention recommendation: Line 36, Line 374, Line 415: I think it is worth adding a caveat/qualifier here about what the end goal of the intervention is. If the goal is to minimise immediate loss of coral cover, then yes, perhaps it is best to prioritise the largest colonies. But, that doesn’t take into account other on-going processes. Firstly, one could argue that the smaller colonies represent younger and more locally adapted/stress tolerant genotypes, and applying intervention efforts across all colonies in a small high-value (heat tolerant)/well connected (larval source)/high-flow (i.e. pathogen source) area might be better than selecting only larger individuals in a broader area using the same amount of time/resources. Secondly, the disease appears to follow contagious transmission dynamics (Muller et al 2020), so if infected small colonies are left untreated in an area around larger colonies that are treated, it may remain a pathogen source to re-infect the larger colonies; I understand this argument led to your original suggestion of culling but that also doesn’t factor in other processes. I would urge you to add a qualifier as such and soften the language because the statement “should be prioritised” is quite strong when these other factors aren’t considered.

- We have modified these statements to modify the language and to include caveats where relevant. Interventions have been proposed as a way to slow down immediate loss of coral cover by aiding in colony survival, but the groups leading these efforts recognize that intervention is not a sustainable approach over the long-term response to the disease outbreak. We have indicated this in the introduction and discussion at L81 and L375, respectively. Additionally, we have added a caveat in the conclusions at L413 stating that intervention efforts on larger colonies should be prioritized when resources are limited.

• Methods: Lines 142-147: The sentences “Statistical tests were run in the R statistical environment” and “non-parametric tests were implemented for all analyses unless otherwise noted”… seemed out of place and premature, because it isn’t clear what tests were run and why? I suggest moving line 146 up to start the section with “to assess variation in disease prevalence among sites and survey times”, and then you can explain the methodological details.

- Changed as suggested

• Figures: It appears that Figures 5a and 6 are the same? Also, Figure 5b could be faceted by site to support the argument that tissue loss didn’t differ among sites.

- Thank you for catching this, this was the result of a small error in the R code used to generate figures. The correct figures have now been included, and Figure 5b has been faceted by site.

---

## [Editor Report · Decision Letter 2]

19 May 2021

Quantifying impacts of stony coral tissue loss disease on corals in Southeast Florida through surveys and 3D photogrammetry

PONE-D-20-25422R2

Dear Dr. Combs,

We’re pleased to inform you that your manuscript has been judged scientifically suitable for publication and will be formally accepted for publication once it meets all outstanding technical requirements.

Kind regards,

Fraser Andrew Januchowski-Hartley, Ph.D.

Academic Editor

PLOS ONE

---

## [Editor Report · Acceptance letter]

18 Jun 2021

PONE-D-20-25422R2 

Quantifying impacts of stony coral tissue loss disease on corals in Southeast Florida through surveys and 3D photogrammetry 

Dear Dr. Combs:

I'm pleased to inform you that your manuscript has been deemed suitable for publication in PLOS ONE. Congratulations! Your manuscript is now with our production department. 

Kind regards, 

on behalf of

Dr. Fraser Andrew Januchowski-Hartley 

Academic Editor

PLOS ONE